# Lightning-induced relativistic electron precipitation from the inner radiation belt

Max Feinland [1] ✉, Lauren W. Blum[2], Robert A. Marshall[1], Longzhi Gan[3], Mykhaylo Shumko[4] & Mark Looper [5]

The Earth's radiation belts are maintained by a number of acceleration, loss and transport mechanisms, and the electron fluxes at any given time are highly variable. Microbursts, which are rapid (sub-second) bursts of energetic electrons entering the atmosphere from the magnetosphere, are one of the key loss mechanisms controlling radiation belt fluxes. Such rapid bursts are typically observed from the outer radiation belt and driven by interactions with whistler mode chorus waves, but they can also occur in the inner belt and slot region, driven by lightning-generated whistlers. This lightning-induced electron precipitation is typically observed at 10s–100s keV, but here we present direct observations of this phenomenon at MeV energies. This unveils a coupling between near-Earth processes, such as lightning, and radiation belt processes, such as relativistic electron microbursts, bridging the gap between Earth weather and space weather.

The Earth is surrounded by toroids of trapped energetic particles, known as the radiation belts. These belts are contained within and shaped by the magnetosphere, which provides protection from the incident solar wind, and shields the Earth from the Sun. The radiation belts, which were one of the first discoveries of the space age[1], consist of two main regions: the inner belt and the outer belt. The outer belt contains very energetic electrons, on the order of MeV[2], while the inner belt typically consists of less energetic electrons, on the order of 10s–100s keV, as well as protons with energies several MeV up to GeV[3,4]. The upper energy limit of electrons in the inner radiation belt is a point of much recent debate, as measurements during solar cycle #24 have shown that there was no significant MeV electron flux in the inner belt between 2012–2015[5,6]. Strong geomagnetic disturbances are needed in order to temporarily populate this region with MeV electrons[2,7].

Radiation belt electron dynamics are highly variable, with many sources of acceleration and loss, causing fluxes to vary by several orders of magnitude. Microbursts are one such loss mechanism, characterized as rapid (sub-second) bursts of energetic electrons precipitating from the magnetosphere into the atmosphere. They are

typically observed from the outer radiation belt, at energies ranging from keV to MeV[8–10], and are often attributed to wave-particle interactions with whistler mode chorus waves[11–13]. 10s–100s keV electron microbursts have also on occasion been observed from the inner radiation belt[14,15]. In this region closer to Earth, lightning-generated whistler waves have been connected to these precipitation bursts, termed lightning-induced electron precipitation (LEP)[14,16]. While LEP of both typical keV electrons and more rare MeV electrons has been indirectly inferred via ground-based very low frequency (VLF) receivers[17,18], the recombination time of the D-region ionosphere means that this method cannot resolve the rapid time evolution of bouncing electron packets. Models as well as the few in-situ electron measurements of lower-energy LEP suggest it can have a repeated periodic signature from the bounce motion of electrons in the magnetosphere[14].

The particles within the radiation belt are trapped in that they typically follow a magnetic field line until the angle between the particle's velocity and the local magnetic field (the local pitch angle, $\alpha$) reaches 90° and the particle "mirrors," or reverses direction, traveling the other way along the field line. This usually repeats indefinitely in

[1]Aerospace Engineering Sciences, University of Colorado, 3775 Discovery Drive, Boulder, CO, USA. [2]Laboratory for Atmospheric and Space Physics, 3665 Discovery Drive, Boulder, CO, USA. [3]Center for Space Physics, Boston University, 725 Commonwealth Ave, Boston, MA, USA. [4]Johns Hopkins University Applied Physics Laboratory, 11100 Johns Hopkins Rd, Laurel, MD, USA. [5]Space Sciences Department, The Aerospace Corporation, 2310 E. El Segundo Blvd, El Segundo, CA, USA. ✉e-mail: max.feinland@colorado.edu

both hemispheres (until the system is disturbed), causing a bouncing motion. Because of this, a subset of all microbursts are so-called "bouncing packets," which are repeated observations of consistently-spaced peaks of electrons, typically of decaying amplitude[9,11]. In general, this phenomenon occurs due to an initial rapid wave-particle interaction that scatters a packet of electrons in pitch angle, some of which immediately precipitate, and some of which mirror or are backscattered back up along the magnetic field line and travel to the magnetic conjugate point in the opposite hemisphere. There, the phenomenon occurs again; some electrons are backscattered and some precipitate. This process can repeat, with more electrons precipitating with each bounce, until the packet has completely precipitated as has been described and modeled by Thorne et al., Chen et al., Cotts et al., Marshall et al., and Peter et al.[11,19–22]. Bouncing microburst packets can be used to determine the spatial extent of a microburst[9,23], but very few have been identified in the literature.

Here we present direct measurements of MeV electron microbursts from the inner radiation belt driven by lightning. In searching for bouncing packets in the SAMPEX/HILT (Solar Anomalous and Magnetospheric Particle EXplorer/Heavy Ion Large Telescope) ≥ 1 MeV electron dataset, measured from a low-altitude (520 × 670 km), high-inclination orbit, we observe a number of bouncing microburst packets in the inner radiation belt that are inconsistent with the chorus generation mechanism. They are unusual due to their highly relativistic (1 MeV, i.e., 0.94$c$ where $c$ is the speed of light) energy at their inner belt location.

## Results

### Properties of LEP events

Forty-five bouncing microburst events were detected at L-shells below 3 over a focused search through a decade of SAMPEX data (detailed in Methods subsection Identification of microbursts). Figure 1 shows examples of these bouncing packet microburst events, where the observed periodicity of the sequential bursts roughly matches the bounce period of 1 MeV electrons at these locations (approximately 200 ms), to within the allowance of the time resolution of the instrument (20 ms). After removing background trends (as shown in Fig. 1, and detailed in the Methods subsection Characterization of event shapes), these bouncing packets exhibit a variety of shapes: about one-third of these events had "crown" (increasing-then-decreasing) profiles (Fig. 1a), another one-third had decreasing profiles (Fig. 1b), and the final one-third was either primarily increasing (Fig. 1c) or another shape entirely (i.e., without notable growth or decay, termed "other"; Fig. 1d).

The properties and locations of these 45 events are displayed in Figs. 2 and 3. Figure 2a shows that they are clustered around $L = 2$ and Fig. 2b shows that they are predominantly observed on the night side of the magnetosphere, consistent with expected LEP properties[14,24]. Figure 2c shows the range of number of peaks identified by our algorithm for each event. These provide a lower limit, as in general, not every peak that seemed obvious by eye was detected by the peak-finding algorithm (e.g., see Fig. 1 and Methods subsection Identification of microbursts). Overall, the detected bouncing packets lasted at least 0.96 to 2.9 seconds. Figure 2d shows the minimum spacing between peaks for each event; the minimum was used because the peak separation is expected to increase in time as the distribution in electron energies produces a spreading of the peaks[23,25]. Assuming a dipole model as described by Schulz and Lanzerotti[26], the characteristic electron energy was calculated for each event. The calculated energies range from 323 keV to 7.81 MeV. However, since the HILT time resolution is 20 ms, and the bounce period around $L = 2$ is approximately 200 ms, there can be quite a bit of uncertainty in the calculated bounce period and therefore electron energy. For example, for an observed bounce period of 200 ms at $L = 2$ and equatorial pitch angle of 15 degrees, the estimated energy would be 1.4 MeV, whereas a bounce period of 220 ms at the same $L$ and pitch angle would be 550 keV. Figure 2e shows the disturbance storm time index (Dst) value

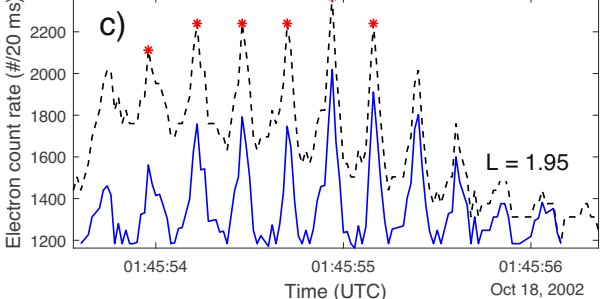

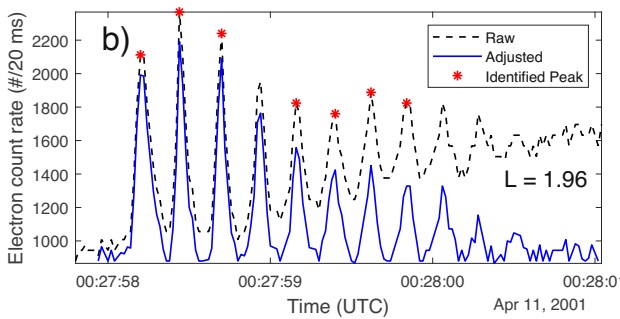

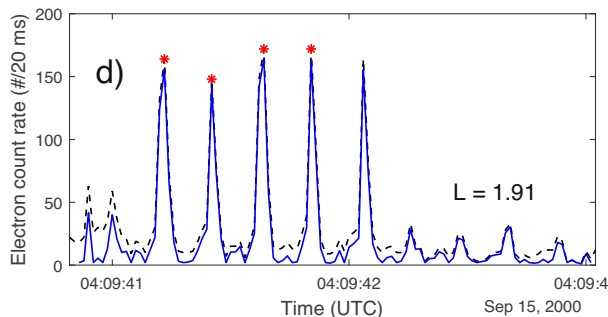

**Fig. 1 | Four example bouncing microburst events of differing shapes. a** A crown-shaped event. **b** A decreasing event. **c** An increasing event. **d** An event identified as other. The dashed black lines indicate the original observations, the red stars indicate peaks as identified by the search algorithm (see "Methods" subsection Identification of microbursts), and the solid blue lines show the background-adjusted observations. The L-shell of each event is included in the bottom right. L-shells describe magnetic field lines, and an L-shell of 2 represents the magnetic field line that is 2 Earth radii away from the center of Earth at its magnetic equator; in three dimensions, this is a "shell." UTC as written in the plot refers to Coordinated Universal Time. Source data are provided as a Source Data file.

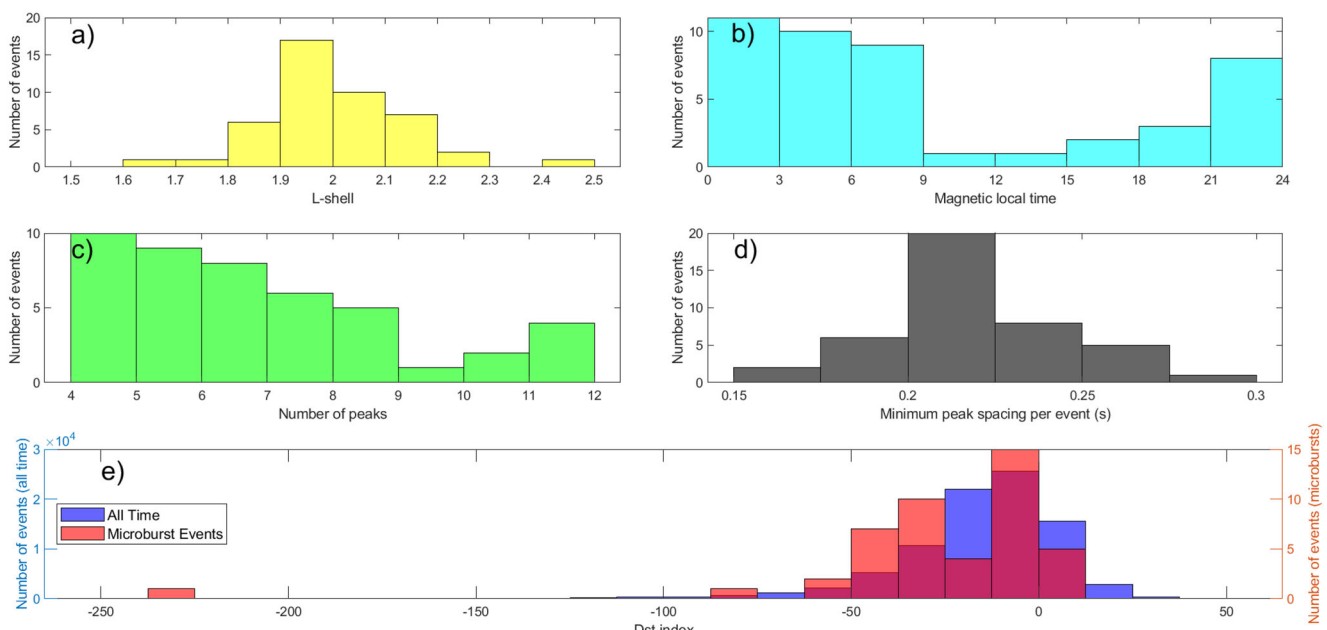

**Fig. 2 | Properties of observed LEP events. a** Distribution in L-shell for each observed packet. The distribution is roughly Normal and centered around $L = 1.99$. **b** Distribution in MLT for each observed packet. There is increased occurrence on the night side of the magnetosphere. **c** Distribution of the number of peaks detected in each observed event. **d** Distribution in minimum observed peak spacings for each event. The distribution is approximately Normal and centered around 0.212s. **e** Distribution in Dst index for microburst events (red), as compared to the Dst index over all time surveyed (blue). The overlap between these quantities is shown in purple. The Dst index during the microburst events is in general lower. Source data are provided as a Source Data file.

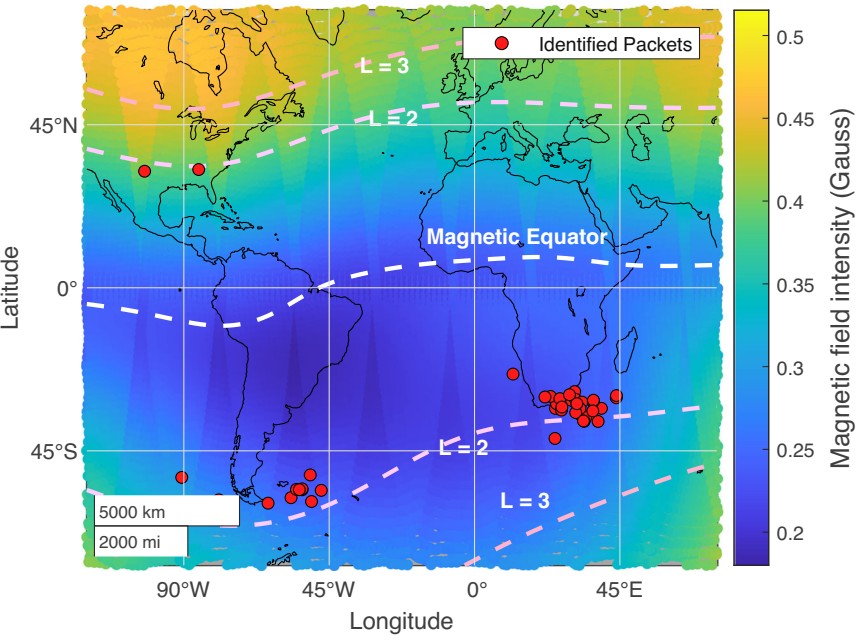

**Fig. 3 | Locations of events.** Geographic locations of each identified bouncing packet (red), with background magnetic field strength shown on the color scale. L-shell contours are superimposed with white dashed lines, calculated using the IGRF-13 model. The red dots indicate the observed events. Source data are provided as a Source Data file.

during these events (dark blue) as compared to the distribution of Dst values over all time throughout the 10 year period. In general, these microburst events tended to occur during geomagnetically disturbed times; the average Dst index during the electron microburst events was lower than the average Dst index during the full 10 years surveyed. Comparisons of Kp and the auroral electrojet (AE) index showed similar trends. (See "Methods" subsection Statistical analysis of Dst index for the statistical significance of this analysis.)

Figure 3 shows the geographic location of all events, revealing a clear clustering around the southern tip of South America, as well as the eastern coast of South Africa, with two events observed in the northern hemisphere above the continental United States. Lightning occurs most often over land mass[27], e.g., North America, Europe, Africa, and Australia, and primarily during the local daytime[28]. However, since ionospheric attenuation is stronger on the dayside, the whistler mode waves that lightning discharges generate can more

easily propagate into the magnetosphere on the nightside[29–31], producing a prevalence of LEP on the nightside. Furthermore, LEP requires a source population of trapped electrons, so LEP events are further restricted to latitudes mapping to the radiation belts. If these microburst events are thought to consist of bouncing packets of electrons, one might expect them to be observed in both hemispheres. However, there is a clear preference for the southern hemisphere in our database of events. One explanation of why SAMPEX primarily observes these events in the southern hemisphere is the asymmetric offset nature of Earth's magnetic field. As shown in Fig. 3, the magnetic field strength at a given L-shell and altitude is weaker in the southern hemisphere compared to the north, thus enabling electrons of a given pitch angle and energy to more easily reach lower altitudes before mirroring in locations within and around the South Atlantic Anomaly. As such, these same bouncing electron packets may be mirroring above the altitude of SAMPEX in the northern hemisphere. While previous studies of LEP at lower energies have observed many events in the northern hemisphere[32], the comparative dearth in our catalog is likely due to the much lower flux of electrons at these higher energies.

Another, more qualitative, feature of these events indicative of LEP is the frequent occurrence of two events observed close in time to each other, on the order of one or fewer minutes apart (see Fig. 4 for an example). These are suggestive of correlated source events (such as lightning strokes occurring during the same storm) causing two microbursts close in time.

### Direct comparison to lightning observations

To further confirm the causes of these MeV microbursts around $L = 2$, we consider the data from the National Lightning Detection Network (NLDN), which contains information about the timing and strength of cloud-to-ground lightning over the United States throughout the time period of these events[33,34]. The lightning coverage is limited to the contiguous United States and some of the Caribbean, as the NLDN was the only readily available dataset whose coverage overlapped with the SAMPEX mission. This coverage limits the number of microburst events in our catalog that can be directly compared to the lightning database. Seven events do occur either with locations or magnetically conjugate locations over the continental US, and of those seven events, three had correlated lightning as determined by the

methodology described in "Methods" subsection Statistical analysis of lightning events. These examples are shown in Fig. 5a–c, where we see a large-amplitude ($\geq 100$ kA[17]) lightning discharge close in location to and within a few seconds of the electron microburst. Based on the lightning-generated whistler and electron propagation times, we expect the LEP event to occur about 1 second after the causative lightning discharge.

If lightning is indeed the cause of the microburst events, one may wonder why all of the U.S. microburst events were not determined to be correlated with lightning. This can be attributed to imperfect lightning detection efficiency, possible in-cloud sources (which are not detected by the NLDN)[35], and the fact that during some of the time period investigated, very large (approximately $\pm 200$ kA) discharges could go undetected due to their complex waveforms[36].

In order to rule out chance coincidence for the correlated events we did find, we calculate the number of events over the contiguous United States with a lightning strike of amplitude $\geq 100$ kA 0–3 seconds before the onset of the event, and compare to the same calculation from the events occurring over Africa. The Americas were shown to have a statistically significant increase in causative lightning when compared with events over Africa (see "Methods" subsection Statistical analysis of lightning events for more detailed information), confirming lightning as a likely driver for events observed over the Americas.

### Comparisons with existing LEP studies

While the global distributions and correlation with NLDN lightning data support the interpretation of these as LEP events, two properties of the observed SAMPEX events differ from previous direct LEP detections: the first being their highly relativistic (MeV) energy, and the second being their shape. To explore the first matter, and to determine when and why particles of these energies might be so close to Earth, Fig. 6a shows the trapped $\geq 1$ MeV electron population, as measured by the SAMPEX HILT instrument, as well as the daily-averaged Dst index and the timing and location of the bouncing microburst packets. In general, a lower Dst index corresponds to a higher degree of geomagnetic activity, and the index can be used to identify geomagnetic storms (stars). From Fig. 6a, we can see that the microburst events tend to follow closely after large dips in Dst. While the inner radiation belt

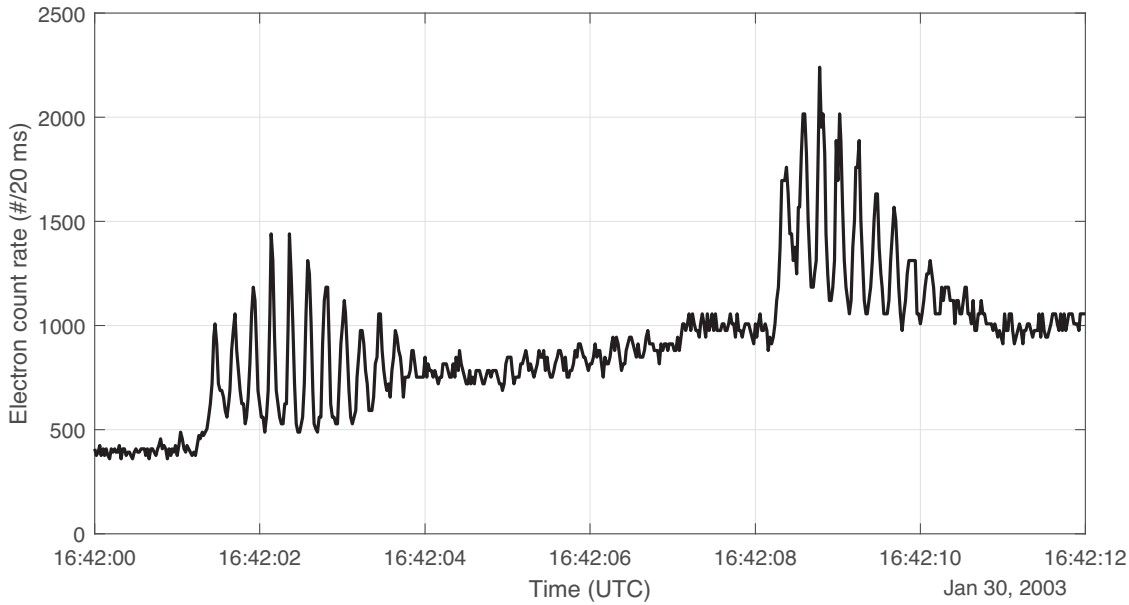

**Fig. 4 | Example related microburst events observed close in time.** There were a number of events like this observed in the catalog, suggestive of two correlated causes (such as two lightning strikes close in time). Source data are provided as a Source Data file.

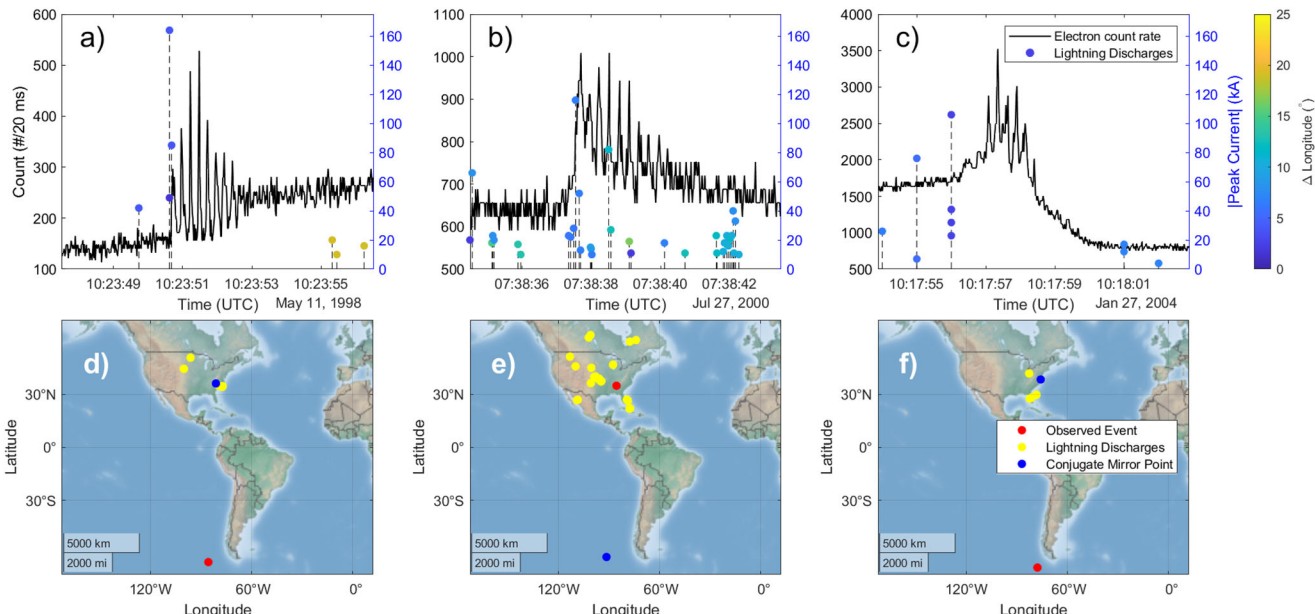

**Fig. 5 | Comparison between microburst events and lightning. a–c** Lightning events overlaid with background-subtracted SAMPEX countrate data. The color bar is the difference in longitude between that of the lightning strike and that of the observed event for the top three panels, such that blue dots indicate discharges close in space to the electron event, and yellow dots indicate far discharges. **d–f** Geographic maps of the location of the lightning events (yellow dots) and electron microburst observations (red dot). The blue dots indicate the magnetically conjugate point of the event, calculated using the IRBEM library. Source data are provided as a Source Data file.

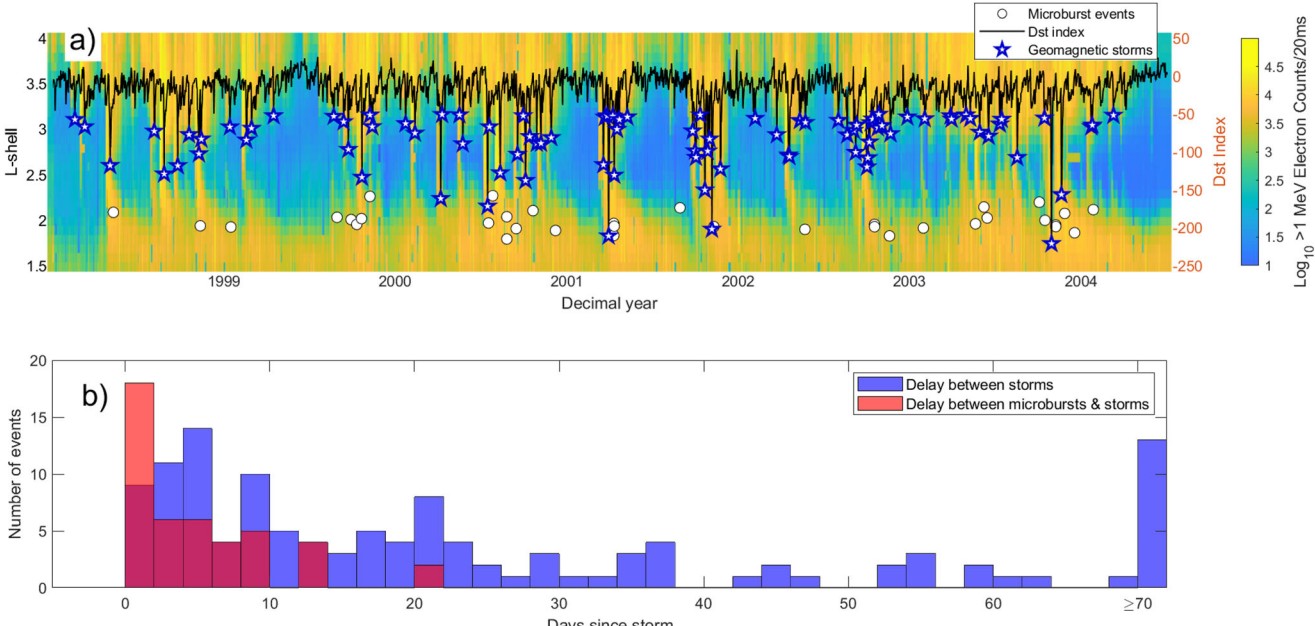

**Fig. 6 | Visualization of geomagnetic storm times over several years. a** Population of ≥1 MeV electrons from 1998 to 2005 as measured by HILT (blue/ yellow colorbar). Overlaid is the daily-averaged Dst index (black line) as well as the observed microburst events (white dots) and geomagnetic storms (blue stars), identified by the Dst index dropping below −50 n T. **b** Histogram of the delay between geomagnetic storms with $Dst_{min} \leq -50$ nT and microburst events (red), overlaid with the spacing between storms (blue). The distribution of delay times between geomagnetic storms and microburst events is markedly shorter on average than the delay between geomagnetic storms. Source data are provided as a Source Data file.

and slot region (i.e., the space between the inner and outer radiation belts) are often devoid of MeV electrons[5,6], during strong geomagnetic activity, energetic particles can get injected into this low L-shell region. Figure 6b quantitatively shows this relationship as well; the distribution of time between geomagnetic storms is much more spread out than that of the time between geomagnetic storms and microburst events, demonstrating that microburst events are more likely to occur during or following periods of increased geomagnetic activity while the slot region was filled with MeV electrons. Geomagnetic storms thus provide the transitory source of MeV electrons at low L-shells, which can then interact with and be scattered by lightning-generated VLF whistlers, producing these MeV microbursts observed around $L = 2$.

Given the relatively frequent occurrence of high-amperage lightning discharges, one may wonder why LEP events were not observed

more often over the 10 year period of examination. There are a few potential explanations for this seeming contradiction, chief among which is the rarity of slot-filling events[5,37,38]. Other factors include the short duration of events and localization of the precipitation patch. Finally, the authored search algorithm was written to be as strict as possible to reduce the likelihood of false positives, and so it is likely that many more of these events occurred in the SAMPEX/HILT data than were identified by this study. Future work could improve the detection efficiency of the search algorithm to locate more events.

Now we turn to the second of the two noteworthy properties mentioned above: the shapes of the microburst events. In the literature, bouncing packets have previously been observed as a decaying signature only[9,14,23]. However, only about one-third of the events reported here showed this temporal signature, with others showing a crown shape or even increasing profile. One explanation for the crown shape could be due to the spatial extent and variability of these events[39]. It is possible that during some of these events, SAMPEX passes through the spatial and temporal peak of electron flux simultaneously, causing the observed crown shape; for example, based on simulation results from Bortnik et al.[39], precipitation patches may vary dramatically over time and space, spanning on the order of 100s of km or fewer up to 1000s of km. The spacecraft may observe a larger spatial variation if it traverses an event near the edge of a precipitation patch as compared with at the center. However, further exploration into the nature of the whistler-electron interactions and spatial variability of such events is needed to confirm the drivers of the variable bouncing packet shapes observed in Fig. 1.

## Discussion

While lightning-induced electron precipitation is a known phenomenon, and LEP of MeV electrons has been inferred from perturbations in subionospheric VLF waves[18], these results constitute the first direct evidence for lightning-driven precipitation of MeV electrons. Furthermore, relativistic electron precipitation by VLF transmitters has previously been observed in this region but only up to 700 keV energies[40,41]. Thus these results may provide insight into the lifetimes and behavior of such high-energy particles so close to the Earth. They also suggest a spatial variability of LEP microbursts previously uncharacterized by former studies; this high temporal resolution, in-situ data allows for an in-depth look at the nature of these events, and reveals details of the wave-particle interaction process that were unable to be captured by previous studies. Extreme lightning bolts, also known as superbolts[42], radiate strong electromagnetic waves[43] and could, in principle, cause stronger LEP than common lightning events. Observations of such lightning discharges and the associated electron precipitation could facilitate better characterization of the nature of these wave-particle interactions.

Accounting for and modeling the behavior of energetic particles in the near-Earth space environment remains an ongoing challenge, particularly for the space industry, where these particles can damage space equipment or even fatally irradiate humans[44]. These results may aid in this challenge and can form the basis for a more detailed analysis of the temporal behavior of MeV electrons in the inner radiation belt. Future high-time resolution measurements of energetic electrons from the inner radiation belts, such as those from the upcoming IMPAX[45] or RADICALS[46] missions, could take advantage of the worldwide lightning networks (which were not readily available during the time surveyed in this study) to provide a more complete, global picture of the relationship between lightning and microbursts in the inner radiation belt. The confluence of Earth-generated processes, such as lightning; near-Earth space phenomena, such as microbursts; and solar-driven activity, such as slot-filling geomagnetic storms; serves as a reminder of the connection between these seemingly unrelated regions.

## Methods

### Identification of microbursts

In order to detect microbursts, data from the Heavy Ion Large Telescope (HILT) instrument on board the Solar Anomalous Magnetospheric and Particle Explorer (SAMPEX) satellite was used. Specifically, the 20 ms count rate portion of the data was searched, over the decade of August 7th, 1996 to August 7th, 2006. Two algorithms were applied using MATLAB, the first of which was adapted from O'Brien et al.[47] to find microbursts, and the second of which was developed by the authors of this paper to identify bouncing packets in particular. The adapted version of the former was implemented as follows:

$$(N_{100} - A_{500})/(1 + \sqrt{A_{500}}) > 5 \qquad (1)$$

where $N_{100}$ is the particle count rate over 100 ms, and $A_{500}$ is the centered running average over 500 ms. The right-hand side of the equation is set to 5 (instead of the original value of 10 in O'Brien et al. [47]) in order to catch lower amplitude events. Any index for which this condition was true was flagged, and if any two flagged indices were fewer than 200 ms apart, they were counted as a part of one interval.

The second algorithm used the intervals identified by the O'Brien algorithm, plus 0.2 seconds at the beginning and 1 second at the end, as inputs for search criteria. It then searched for peaks within the interval (i.e., local maxima; any data point that is higher than both of its neighboring points). This was implemented using MATLAB's function findpeaks, available from the Signal Processing Toolbox. In order to qualify as a peak, the prominence of the peaks was required to be at least 0.25 times the total range of count rate values spanned by the input interval. In essence, the O'Brien algorithm was used as a candidate finder so that a prominence requirement could more efficiently be imposed. In order to minimize the number of false peaks found, a minimum peak separation of at least 0.06 seconds was also applied; that is, if two peaks were identified that were fewer than 0.06 seconds apart, the algorithm ignored the smaller peak and only counted the larger one. This function searched for runs of at least four consecutive peaks, and flagged the interval as a candidate event if the peaks were consistently spaced (defined as the difference in spacing being less than 0.15 times the mean spacing), and discarded intervals that did not meet this criterion. After finding these intervals, the algorithm checked that a) the total time interval was less than 15 seconds long, and b) there were at least 10 unique count rate values recorded over the course of the interval. The first condition exists to focus in on only very short, bursty instances; the second removes periods during very high count rates where the quantization of the countrates compressed for inclusion in telemetry is coarse, and thus oscillates between two or three values, creating false peaks.

Since these two scripts were written for more general microburst/ bouncing packet identification, an extra criterion was added to narrow down the scope of the study to the inner radiation belt and slot region only: the L-shell, as determined from SAMPEX ephemeris data, must be equal to or lower than 3.

Over the course of the time period surveyed, 68 events were detected by these algorithms. Of those, 45 showed clear bouncing packets by eye, and were included in this study (some events removed were noisy or contained false positives with very wide peaks whose exact tip could be at multiple locations, creating difficulty in determining the spacing between peaks). The identification process was designed to be as strict as possible to decrease the number of false positives, meaning that some events were likely missed.

### Characterization of event shapes

In addition to identifying bouncing microburst packets, we also wanted to characterize their magnitude and shape, and thus implemented a procedure in MATLAB to remove more slowly varying background trends from the microburst events (e.g., Blum et al.,

Douma et al., O'Brien et al.[8,48,49]). This involved finding the local minima of each interval, and then interpolating between these points to capture the behavior of the background activity. The background activity was then subtracted from the interval to show the underlying shape. The shapes of each event were then classified based on these background-subtracted profiles.

## Statistical analysis of Dst index

A two-population t-test was performed in MATLAB using the function test, available from the Statistics and Machine Learning Toolbox, between the hourly Dst index during the entire epoch surveyed (1 decade) and the Dst index during the electron microburst events. The likelihood that these two populations were statistically homogeneous was $p = 0.0046$. Using the standard significance level of 0.05, we are able to reject the null hypothesis and conclude that the microburst events occurred during statistically lower Dst indices (corresponding to higher geomagnetic activity). The mean Dst index during microburst events was −26.73 nT, while the mean Dst for all time surveyed was −16.40 nT.

## Statistical analysis of lightning events

We present a statistical analysis, performed in MATLAB, between the events with actual or conjugate mirror points over the contiguous United States and those observed over Africa, to determine the likelihood of chance coincidence being the cause of the correlation between lightning discharges and electron microbursts during some of our events. Events over the Atlantic Ocean were omitted due to the NLDN's imperfect detection efficiency in this region; its coverage should extend over the entirety of the United States, and there should be no coverage over Africa, whereas the coverage over the Atlantic cannot be definitively qualified in this way. If there was a lightning strike with a peak current larger than ±100 kA[17] between 0 and 3 seconds before the onset of the microburst event, this was counted as a success, whereas if there was no such event, it was a failure. The proportion of successes for each population was compared using a two-population t-test, which returned a $p$-value of $1.16 \times 10^{-4}$, or a 0.012% chance of these results assuming the populations are homogeneous. Using the standard significance level of 0.05, we are able to reject the null hypothesis and conclude that these populations are in fact different. The microburst events over the United States have a much higher rate of correlated lightning, and we can therefore ascertain that these events are not just coincidentally aligned with lightning activity and are thus indeed the result of lightning-induced electron precipitation.

## Data availability

The processed SAMPEX data used for the identification and characterization of microbursts in the study are available from the SAMPEX Data Center at https://izw1.caltech.edu/sampex/DataCenter/DATA/HILThires/State4/ and https://izw1.caltech.edu/sampex/DataCenter/DATA/PSSet/Text/ (count rate and attitude/ephemeris data, respectively) which are accessible to the public. The OMNI dataset, used to pull Dst data, is available from the NASA SPDF OMNIWeb interface at http://omniweb.gsfc.nasa.gov which is accessible to the public. The datasets generated during and analyzed during the current study are available from the corresponding author upon request. The raw Vaisala lightning data are protected and are not available due to data privacy laws. The truncated version of this data is available from the corresponding author upon request. The data used to generate all plots shown in this study are provided in the Source Data file. Source data are provided with this paper.

## Code availability

The computer code used to identify the bouncing microburst packets is available from Code Ocean[50], which is available to the public. The O'Brien microburst algorithm as originally implemented is detailed in

O'Brien et al.[47], and the adapted version as used in this study is available within the driver script specified above.

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

## Acknowledgements

We acknowledge the use of the IRBEM library (version 5.0.0), the latest version of which can be found at https://doi.org/10.5281/zenodo.6867552. We sincerely thank Dr. Ryan Said from Vaisala, Inc. for providing lightning data from the NLDN. This work was supported in part by NASA H-SR grant #80NSSC21K1682 (LB) and NSF grant #AGS2123253 (LB).

## Author contributions

M.F. wrote the manuscript and search algorithm and generated the plots under the supervision of L.B. L.B. edited the manuscript and coordinated the collaboration. R.A.M. coordinated the acquisition of lightning data and provided valuable insight through discussion into LEP behavior. M.S. and L.G. provided valuable input through discussion and interpretation of observations, as well as confirmation of bounce period calculations. M.L. offered engineering insight into the workings of the HILT instrument. All authors reviewed and provided feedback on the manuscript.

## Competing interests

The authors declare no competing interests.
