## [Peer Review File · Nature Communications]

Lightning-induced relativistic electron precipitation from the inner radiation belt

Editorial Note: Parts of this peer review file have been redacted as indicated to avoid any copy right infringement.REVIEWER COMMENTS

Reviewer #1 (Remarks to the Author):

Review report by Ondrej Santolik

on Manuscript#: NCOMMS-23-62997-T "Lightning-induced relativistic electron precipitation from the inner radiation belt"

submitted to Nature Communications by Max Feinland et al.

The paper represents important advances. It shows significant direct measurements of lightning induced electron precipitation at MeV energies in a form of bouncing microburst sequences. The data sources are limited to the measurements of the SAMPEX spacecraft between 1996 and 2006, with lightning detection data from NLDN. No measurements of whistlers or atmospherics are included but the paper still contains enough new evidence.

I appreciate that the submission is very carefully prepared. The text is well written and easy to read, taking care of explications for non-specialists. The figures are clean and well illustrate the results. I only have a set of minor specific recommendations for improving the text and figures (see below).

In my opinion, this work is a valuable contribution to the existing literature on lightning induced electron precipitation. I recommend its publication in Nature Communications after my comments are considered by the authors.

Specific comments:

line 59 It would be fair to state already here (and not only later on line 175) that Inan et al 1988 in ref 15 also found - though indirectly - LEP at MeV energies from the inner belt.

lines 72-73 This mechanism is plausible but requires a better description of the process of "losing more and more electrons with each bounce" following the initial pitch angle scattering event. I suggest the authors to add a brief description of the process. This is important for explication of the shapes of the bounce sequences.

lines 84-85 This looks like an unnecessarily uncertain statement to me: as the intervals between the peaks seem to be approximately constant for each observation in Fig. 1, it should be possible to derive the median/mean value of the bounce travel time, and consequently the characteristic electron energy in each of the 45 events, assuming the hypothesis from line 72. The summary results can be then plotted in a histogram of bounce delays and a histogram of energies, together with the histograms in Fig. 2, without much increasing the length of the paper.

line 126 It would be interesting to know if any of the two bouncing microburst events detected over the continental US has its source lightning in the NLDN data. I guess that its timing might be closer to the first peak of the bouncing microburst event compared to the events in the Southern hemisphere. In any case I suggest the authors to show also the third case of lightning correlated to bouncing microbursts, either in the paper or in the supporting material.

line 135 The attached file with a list of NLDN detections for the two events from Fig. 2 indicates observations of several multi-stroke lightning discharges. It would be good to mention this fact in the text. Also the timing in this supporting file is limited to entire seconds so it is impossible to use it for ruling out the possibility that each peak in the electron flux corresponds to a separate stroke. Fig. 4 seems to use a better timing precision so it should be easy to rule out that hypothesis.

line 145 See my comment to line 59

line 158-161 (also 240-241) It would be nice to see here a brief discussion of the very low occurrence of the detected LEP events. Even during the geomagnetically disturbed times, when the MeV electrons move to lower L shells, there must be significantly more >100 kA

discharges than the number of LEP events reported in the paper. The low probability of encountering localized events (see also my comment for line 169) might compensate it. Rough quantitative estimates would be useful here. Maybe a large part of events which would be encountered along the spacecraft orbit is missed by the detection algorithm. Interestingly, the lack of detections might also have a natural cause, and additional conditions may play a significant role.

line 169 Although the need of further exploration is stated, it would be nice to see a better description of the hypothesis to explain the observed shapes. Do I guess correctly that this hypothesis is based on an assumption that electron fluxes for all the events always decrease in time at a fixed place, while the spatial distribution of electron fluxes at fixed time is very narrow around a central position ? (or just narrow in L-shell, and not not in MLT ?) The spacecraft then would scan this spatial distribution as well as the temporal evolution of the event. The increases of fluxes can therefore only be due to spatial variations and they typically seem to happen on the time scale of ~1 second, this would probably results in a spatial scale of LEP events on the order of several hundred meters at the spacecraft altitude. I recommend the authors to add a brief discussion of these or similar straightforward inferences, and put them in the context of existing literature.

Reviewer #2 (Remarks to the Author):

This is a very interesting paper, clearly written and well articulating its arguments. The argument of SAMPEX detection of the Lightning-induced Precipitation Bursts of MeV electrons seems solid and well justified. I am very pleased to see this paper, as for many years during the lifetime of SAMPEX, Bernie Blake and myself have talked extensively about the detection efficiency and high-time resolution of the SAMPEX detectors as being so very well suited for detection of LEPs.

In the second paragraph, Lines 54 through 60, there needs to be a reference to the only two papers that reported direct low altitude satellite measurements of LEP events, strongly correlated with electromagnetic waves (whistlers) from lightning:

Inan, U. S., D. Piddychiy, W. B. Peter, J. A. Sauvaud, M. Parrot (2007), DEMETER satellite observations of lightning-induced electron precipitation, *Geophys. Res. Lett.*, 34, L07103, doi:10.1029/2006GL029238.

Voss, H. D., et al. (1984), Lightning-induced electron precipitation, *Nature*, 312, 740–742, doi:10.1038/312740a0.

In addition, in the context of the discussion of bouncing packets of electrons, and the asymmetry of the northern and southern mirror heights, there needs to be a reference to the paper that analyzed LEP events observed in the northern hemisphere in great detail:

219. Inan, U. S., M. Walt, H. Voss, and W. Imhof, Energy Spectra and Pitch Angle Distribution of Lightning-Induced Electron Precipitation: Analysis of an Event Observed on the S81-1 (SEEP) Satellite, *J. Geophys. Res.*, 94, pp. 1379-1401, 1989.

I strongly recommend the paper for publication with the above additions.

March 13 2024

Object: Review of NCOMMS-23-62997-T

The article written by Feinland et al. entitled "*Lightning-induced relativistic electron precipitation from the inner radiation belt*" which has been submitted to Nature Communications shows the precipitation of MeV electrons caused by lightning-generated whistlers. As written by the authors, this result is new and important. The demonstration is made by direct satellite measurements as the authors have found rare events of microbursts in the SAMPEX satellite measurements database. Microbursts, as well defined in the article, are high-temporal flux variations recorded at the fast rate of 20 μ s. These measurements allow to see the fine temporal structures of the precipitating flux which is very interesting, with application to modeling. These structures are shown and discussed. The demonstration of the association with lightning-generated electromagnetic waves is well conducted to my point of view. First, 68 events are identified over 10 years of measurements, which is rare but still significant for the demonstration. Among them 45 which exhibit clear microburst signals are selected. Each of the 45 events is associated with actual lightning events recorded by the National Lightning Detection Network (NLDN), which contains the timing and strength of cloud-to-ground lightning over the US. I think the demonstration stands, made in a convincing manner. The article is very well written as well, which makes it pleasant to read. The Nature group format is respected, so that all concepts are well defined, without jargoning, and the article is made to reach the largest audience as possible. This is a success.

Nevertheless, I will recommend rejection of this article, with encouragement to resubmit, once the major corrections I am asking below are made. There are two major sources of disagreement with me in this article, each related to figure 5 and figure 6.

Briefly, figure 5 shows the long-term dynamic of >2 MeV electrons correlated with the electromagnetic lightning waves induced microburst precipitations. But the demonstration of the link of MeV electron precipitations with lightning waves is made for >1 MeV electrons in the article, certainly not for >2 MeV. This is totally misleading and not acceptable. I ask Fig 5 to be made for >1 MeV electrons, though I am finding a caveat with the >1 MeV SAMPEX channel reported in Selesnick et al. JGR 2015, which may prevent that approach to be a solution. Most of the data collected at energy threshold of 1 MeV are caused by lower energy electrons or high-energy protons (Selesnick et al. JGR 2015). As there is an issue with the >1 MeV data, the authors should say something about it. Also, there is a second issue in this figure made from 10 years of measurements show geomagnetic activity in a very coarse way (1-month averaging). The events are plotted on top of the flux but cannot be well related to geomagnetic activity, which is not well visible at this scale. This second aspect is also misleading. That issue will be removed by a series of new distributions which I ask below the authors provide in order to properly relate their events o geomagnetic activity. Maybe >1 MeV flux can be replaced by other types of information (as slot filling event identification or the distributions I propose or other ideas I give below) or this figure should be removed.

The second disagreement is about Figure 6, in which we face here a coarse sketch of a dynamics which brings absolutely nothing to the article except a too caricatural, sometimes wrong, view of the radiation belt dynamics and of their projection onto the Earth's map (fig 6a-b-c) as well as onto the projection of the Earth magnetic field (fig 6d). This is way too naïve. Not in Nature under my watch. The text associated with this figure is totally minor (copied below for your information). I simply recommend the removal of the text and the figure. This will not diminish the scientific achievements, which are mostly contained in Figure 1 to 4.

More corrections follow but these two reasons above are the reasons of my current rejection. I am not sure corrections can be done in a very short time as sometimes asked by Nature, particularly the correction about Figure 5. I recommend to give the authors the time they need to properly address them. Also, two new satellite missions will be in orbit in a couple of years or so to study more microburst events. I ask the authors to mention them in the perspectives as their article can be inspiring and directly influence and impact new research studies. After appropriate corrections, I think this article will have the high potential and results to be considered for publication in Nature Communications.

Corrections/comments follow

Various general comments

The rarity of these event in the inner belt is linked to the absence of these MeV electrons in great majority. Please refer to Fennell et al. GRL 2015 on the upper limit of RB electrons and Baker et al. Nature 2015 on the "barrier" prohibiting MeV electrons to reach the inner belt. Please cite also, the rare intrusion of MeV electrons in the inner belt as discussed in Claudepierre et al. JGR 2019. These three references seem to me mandatory.

Abstract: "...where they do not reside long-term". Unclear. Do you speak of the electrons seen in the microburst because they vanished after a short term or do you speak in general of radiation belt electrons? In the first case, you are not discussing the lifetime of these electrons; could you? Please rephrase to be clear.

"these results constitute the first direct evidence for lightning-driven precipitation of MeV electrons, illuminating a new perspective on the lifetimes and behavior of such high-energy particles so close to the Earth.": this is indeed a very good result worth publishing in Nat. Comms. It is not only by lightning waves but by any VLF waves. I would like you mention that Cunningham et al. GRL 2020 could show precipitations of relativistic electrons by VLF-Transmitter waves up to 600-700 keV only, still above 450 keV found by Sauvaud et al. (2008), but not above 700 keV. The article is the first one to reach >1 MeV. That should support your claim even more.

Please indicate in the text what is the altitude of SAMPEX.

Additional information in Fig 2.

Please give the following distributions:

- Distribution in duration of the events
 - Distribution in Nb of peaks of the events (if interesting and different from the duration, if not, then just consider a quick comment on this number).
 - Distribution with geomagnetic activity, maybe Kp, AE and Dst. Do you need to introduce “*” (starred indices) i.e the average over the previous 24h as sometimes done? Please choose according to the relevance you find.
 - Any other idea? See below the connection with slot filling events and how you may establish a correlation with them, shown by a distribution
- Please try to enrich Fig 2 with the relevant distributions you could find. Please make fig 2 more meaningful and impactful. (As such it is great but a bit limited and we want to know more, plus add the information which is currently in Fig. 5, which cannot stay if showing >2MeV).

About Fig. 5 and related text

Fig5 shows >2MeV electrons while here the authors use >1 MeV electrons. This is not possible because it suggests to the reader the precipitation are occurring at >2MeV, way above the natural limit of inner belt electrons. Fig 5 should plot the energy channel which is used in the article and not a different one. Now, it is unfortunate that the main >1 MeV channel is likely not usable as found by Selesnick in 2015.... I do not think this issue can be put under the rug. The corruption of the >1 MeV channel is though –I think—fully unrelated with the microburst detector. Please discuss that once the results of Selesnick et al. 2015 will be cited.

Figure 5, which is made from 10 years of measurements, shows geomagnetic activity in a very compressed way due to the packing of the years. The resolution of Dst is very coarse: an averaged value over 1 month. The events are plotted on top of the flux but cannot be well related to local geomagnetic activity, which is not well visible at this scale. This second aspect is also misleading. I think the distribution with Dst I asked above will be enough to relate one with the other, certainly more precise.

About the text related to figure 5:

-“ From the figure, we can see that the microburst events tend to follow closely after large dips in Dst.”. I cannot conclude that from the fig.5 due to the Dst resolution.

-“ Figure 5 shows that many of the microburst events occurred during or following periods of increased geomagnetic activity, while the slot region was filled with MeV electrons.”. I do not see that from the figure.

Can you make a better demonstration? Can you relate or correlate slot filling events with these events? See Turner et al. 2015. That may work. But this will only say that the presence of these electrons in the inner belt is rare enough that they have to be brought during active time, which I do not contest. As the authors, I think high activity is required to have the presence of these electrons in the inner belt in the first place. But, then, what is happening? Do we have the situation that any of the many lightning waves will cause their instantaneous or rapid precipitations? Or, these electrons stay some time trapped until they are hit by a lightning wave, totally independently of the geomagnetic activity? The authors are suggesting the first case based on a crude comparison I contest. Please make a better work at relating activity with these events. For instance, what is the time between the event and the closest significant storm which you think brought these electrons? ... how would be that

distribution for the 45 events? As you do, you may find a better way to demonstrate your claim, without the use of the current fig. 5.

About Fig. 6 and related text

I am simply against Fig 6 by all means. In a-b-c the dynamics is caricatural. In fig 6d, the spread of the radiation belts is way too rough. This figure is an introduction figure to maybe a course and even I found it too inaccurate to show it to students. The SAA is giant, the radiation belts are too wide for the North one particularly, etc. If drawn by Picasso, maybe.... There is no particular message in this figure. The storm makes a relativistic radiation belt appearing in c) gone in a) and b). Where did you see that? This figure is too speculative and naïve. I am recommending not to publish that figure. The article is perfectly fine without. Looking at the text referring to this figure in the concluding paragraph, copied below and colored in blue, it is very weak and can simply be removed:

“The combination of numerous disparate phenomena are needed to produce the spatial and temporal distributions of MeV LEP events revealed here. Figure 6 highlights this combination, with Figures 6a-c demonstrating the dependence in time (i.e. during or after geomagnetic storms), and Figure 6d showing the dependence in space (i.e. the overlapping of the inner radiation belts, lightning distributions, and the South Atlantic Anomaly). Accounting for and modelling the behavior of energetic particles in the near-Earth space environment remains an ongoing challenge, particularly for the space industry, where these particles can damage space equipment or even fatally irradiate humans [30]. These results may aid in this challenge, and can form the basis for more detailed analysis into the temporal behavior of MeV electrons in the inner radiation belt. In addition, the confluence of Earth-generated processes, such as lightning; near-Earth space phenomena, such as microbursts; and solar activity, such as slot-filling geomagnetic storms; serves as a reminder of the connection between these seemingly unrelated regions.”

With the removal I ask, this last paragraph would become perfectly fine with me:

“Accounting for and modelling the behavior of energetic particles in the near-Earth space environment remains an ongoing challenge, particularly for the space industry, where these particles can damage space equipment or even fatally irradiate humans [30]. These results may aid in this challenge, and can form the basis for more detailed analysis into the temporal behavior of MeV electrons in the inner radiation belt. In addition, the confluence of Earth-generated processes, such as lightning; near-Earth space phenomena, such as microbursts; and solar activity, such as slot-filling geomagnetic storms; serves as a reminder of the connection between these seemingly unrelated regions.”

That showed how minor that text was. See below where I propose the authors more perspectives.

Gaining impact in the perspectives

Lightning strikes can be tracked with the WWLLN lightning detection network as explained in the text. As indicated by the authors, at the time their data were collected, this network was

unavailable. This causes the authors to limit their analysis in the US region due to the NLDN network coverage, which accuracy is also lower over the Atlantic region, where the authors have many events. This is an unavoidable caveat. However, for new upcoming missions, which are also targeting microburst events, such as IMPAX (led by Minnesota Univ.) and the Canadian RADICALS missions, this network should provide a good way to identify accurately the location of the lightning source of lightning related microbursts as well as enlarge the region of study to the whole world, instead of the US only. Your article is an invitation to do so. Therefore, I would like you take advantage of this argument to make your article more impactful in directly citing in the perspectives the possibility of extending your results with the upcoming new measurements from IMPAX or RADICALS, themselves being possibly related to new lightning network such as WWLLN (see for instance in Ripoll et al. GRL 2019) for a world-wide assessment.

Minor corrections

In Fig 4, try neon yellow for lightning, not blue because one lightning in the Florida see is not visible.

Fig 6a: “an asymmetrical distribution of energetic particles about the Earth”. Rephrase.
Fig6d: not commented in the legend. Though I recommend to remove that figure.

Please change

“ lightning discharges generate can more easily propagate into the magnetosphere on the nightside, producing a prevalence of LEP on the nightside only [21–23].”

By

“lightning discharges generate can more easily propagate into the magnetosphere on the 102 nightside [21–23], producing a prevalence of LEP on the nightside only.”

(unless these articles have a specific aspect on LEP but I don't think so. Still, if so, you can break the citation in two pieces, one relative to wave observations, the other to LEP)

References

Baker, D. N., Jaynes, A. N., Hoxie, V. C., Thorne, R. M., Foster, J. C., Li, X., et al. (2014). An impenetrable barrier to ultrarelativistic electrons in the Van Allen radiation belts. *Nature*, 515(7528), 531–534. <https://doi.org/10.1038/nature13956>

Claudepierre, S. G., O'Brien, T. P., Looper, M. D., Blake, J. B., Fennell, J. F., Roeder, J. L., et al. (2019). A revised look at relativistic electrons in the Earth's inner radiation zone and slot region. *Journal of Geophysical Research: Space Physics*, 124, 934–951. <https://doi.org/10.1029/2018JA026349>

Cunningham, G. S., Botek, E., Pierrard, V., Cully, C., & Ripoll, J.-F. (2020). Observation of high-energy electrons precipitated by NWC transmitter from PROBA-V low-Earth orbit satellite. *Geophysical Research Letters*, 47, e2020GL089077. <https://doi.org/10.1029/2020GL089077>

Fennell, J. F., S. G. Claudepierre, J. B. Blake, T. P. O'Brien, J. H. Clemmons, D. N. Baker, H. E. Spence, and G. D. Reeves (2015), Van Allen Probes show that the inner radiation zone contains no MeV electrons: ECT/MagEIS data, *Geophys. Res. Lett.*, 42, 1283–1289, doi:10.1002/2014GL062874.

Ripoll, J.-F., Farges, T., Lay, E. H., & Cunningham, G. S. (2019). Local and statistical maps of lightning-generated wave power density estimated at the Van Allen Probes footprints from the World-Wide Lightning Location Network database. *Geophysical Research Letters*, 46. <https://doi.org/10.1029/2018GL081146>

Sauvaud, J.-A., R. Maggiolo, C. Jacquy, M. Parrot, J.-J. Berthelier, R. J. Gamble, and C. J. Rodger (2008), Radiation belt electron precipitation due to VLF transmitters: Satellite observations, *Geophys. Res. Lett.*, 35, L09101, doi:10.1029/2008GL033194.

Turner, D. L., et al. (2015), Energetic electron injections deep into the inner magnetosphere associated with substorm activity, *Geophys. Res. Lett.*, 42, doi:10.1002/2015GL063225.

Author Responses to Reviewer Comments

“Lightning-induced relativistic electron precipitation from the inner radiation belt”

Manuscript submitted to Nature Communications by Feinland et al.

We thank the reviewers for their thoughtful contributions and suggestions and we have addressed each comment below.

Reviewer #1

- **R1C1:** line 59 It would be fair to state already here (and not only later on line 175) that Inan et al 1988 in ref 15 also found - though indirectly - LEP at MeV energies from the inner belt.
 - **Response:** Thank you for this comment, this was incorporated.
- **R1C2:** lines 72-73 This mechanism is plausible but requires a better description of the process of "losing more and more electrons with each bounce" following the initial pitch angle scattering event. I suggest the authors to add a brief description of the process. This is important for explication of the shapes of the bounce sequences.
 - **Response:** To clarify this process, the description was changed as follows: “In general, this phenomenon occurs due to an initial rapid wave-particle interaction that scatters a packet of electrons in pitch angle, some of which immediately precipitate, and some of which mirror or are backscattered back up along the magnetic field line and travel to the magnetic conjugate point in the opposite hemisphere. There, the phenomenon occurs again; some electrons are backscattered and some precipitate. This process can repeat, with more electrons precipitating with each bounce, until the packet has completely precipitated as has been described and modeled by Thorne et al. 2003, Chen et al. 2020, Cotts et al. 2011, Marshall et al. 2018, and Peter et al. 2004.” We hope that this description is clearer.
- **R1C3:** lines 84-85 This looks like an unnecessarily uncertain statement to me: as the intervals between the peaks seem to be approximately constant for each observation in Fig.1, it should be possible to derive the median/mean value of the bounce travel time, and consequently the characteristic electron energy in each of the 45 events, assuming

the hypothesis from line 72. The summary results can be then plotted in a histogram of bounce delays and a histogram of energies, together with the histograms in Fig.2, without much increasing the length of the paper.

- **Response:** Thank you for this comment. Since the SAMPEX/HILT time resolution is 20 ms, and the expected bounce period around $L = 2$ is approximately 200ms, there can be quite a bit of error in the calculated bounce period. For example, for an observed bounce period of 200 ms at $L = 2$ and equatorial pitch angle of 15 degrees, the estimated energy would be 1.4 MeV, whereas a bounce period of 220 ms at the same L and pitch angle would be 550 keV. The top-left figure below highlights the errors inherent in the bounce period calculation; not all algorithm-identified peaks are at the locations that a human might place them. Furthermore, as HILT provides an integral energy channel, there is likely some range of energies producing the signature, resulting in the peaks spreading in time due to energy dispersion (as in Figure 1 of Shumko et al. GRL 2018, last figure in this response), introducing yet more error into the calculation. Following your suggestion, an example event is shown below. While the observed bounce period is in general agreement with the theoretical estimate for a 1MeV electron, due to the time resolution of the instrument this is generally to within +/- 20ms accuracy. A more detailed analysis of the observed bounce periods, with precise peak fitting and finding and greater quantification of error, could be interesting as a follow-on study. We have now modified the text to clarify why the original statement included this uncertainty.

➤ **R1C4:** line 126 It would be interesting to know if any of the two bouncing microburst events detected over the continental US has its source lightning in the NLDN data. I guess that its timing might be closer to the first peak of the bouncing microburst event compared to the events in the Southern hemisphere. In any case I suggest the authors to show also the third case of lightning correlated to bouncing microbursts, either in the paper or in the supporting material.

- **Response:** Thank you for this suggestion, we have added the third correlated event into Figure 5, which happened to be one of the observed events over the United States and which indeed appears to initiate closer in time to the source lightning.
- The other US event looks like so:

As noted in the paper, there was a period of time (from roughly 2002 to 2006) during which very large-current lightning discharges were scrubbed from the data due to the nature of the quality control algorithm that was used. Given that this

microburst event occurred during this time, and there are numerous lightning strikes that are close in longitude surrounding the microburst event, we consider it likely that there was a causative lightning discharge of larger magnitude close to the microburst onset that may have been removed from the dataset.

- **R1C5:** line 135 The attached file with a list of NLDN detections for the two events from Fig. 2 indicates observations of several multi-stroke lightning discharges. It would be good to mention this fact in the text. Also the timing in this supporting file is limited to entire seconds so it is impossible to use it for ruling out the possibility that each peak in the electron flux corresponds to a separate stroke. Fig. 4 seems to use a better timing precision so it should be easy to rule out that hypothesis.
 - **Response:** There are indeed frequently multiple lightning discharges surrounding any one LEP event, and there were also frequently observed “sister” events, or bouncing microburst packet events observed within one minute of each other (see example below), further supporting the LEP hypothesis. To acknowledge this, we have added the following text to the manuscript: **“Another, more qualitative, feature of these events indicative of LEP is the frequency with which two events were observed close in time to each other, on the order of one or fewer minutes apart. These are suggestive of correlated source events (such as lightning strokes occurring during the same storm) causing two microbursts close in time.”**
 - Because the lightning data was from Vaisala, it was proprietary and its distribution was restricted to certain use cases. One of these was the “truncation” of the data, i.e. the removal of extra decimal places and unnecessary parameters that were included in the source file and omitted in the submitted version. The full data does contain timestamps to the nearest nanosecond. For this reason, it is generally possible to rule out the possibility that each peak corresponds to a different discharge.

- **R1C6:** line 145 See my comment to line 59
 - **Response:** Added “direct” to “LEP studies” to distinguish from indirect VLF methods.

- **R1C7:** line 158-161 (also 240-241) It would be nice to see here a brief discussion of the very low occurrence of the detected LEP events. Even during the geomagnetically disturbed times, when the MeV electrons move to lower L shells, there must be significantly more >100 kA discharges than the number of LEP events reported in the paper. The low probability of encountering localized events (see also my comment for line 169) might compensate it. Rough quantitative estimates would be useful here. Maybe a large part of events which would be encountered along the spacecraft orbit is missed by the detection algorithm. Interestingly, the lack of detections might also have a natural cause, and additional conditions may play a significant role.
 - **Response:** The authors agree on all suggestions; it is likely that the low time density of events is related to their short durations, localized nature, and the specific (and rare) conditions needed to produce the MeV LEP. The authors also agree that the detection efficiency is imperfect; the detection algorithm was designed to be as strict as possible to reduce the likelihood of false positives. This means, however, that many microburst events go undetected with the current methodology. For example, there were occasionally “sister” events, or bouncing microburst packet events observed within one minute of each other (see previous example), where one event was identified by the algorithm but another was not (only by eye). As such, we have added the following text:

- “Given the relatively frequent occurrence of high-peak-current lightning discharges, one may wonder why LEP events were not observed more often over such a long epoch. There are a few potential explanations for this seeming contradiction, chief among which is the rarity of slot-filling events [Fennell, Baker, Claudepierre]. Other factors include the localization of the precipitation patch and the short timescale over which the precipitation patch propagates. Finally, the authored search algorithm was written to be as strict as possible to reduce the likelihood of false positives, and so it is likely that many more of these events occurred in the SAMPEX/HILT data than were identified for this study.”
 - The example figure, for which the second packet was detected but the first was not is shown in R1C5, highlighting the imperfect detection efficiency.
- **R1C8:** line 169 Although the need of further exploration is stated, it would be nice to see a better description of the hypothesis to explain the observed shapes. Do I guess correctly that this hypothesis is based on an assumption that electron fluxes for all the events always decrease in time at a fixed place, while the spatial distribution of electron fluxes at fixed time is very narrow around a central position ? (or just narrow in L-shell, and not not in MLT ?) The spacecraft then would scan this spatial distribution as well as the temporal evolution of the event. The increases of fluxes can therefore only be due to spatial variations and they typically seem to happen on the time scale of ~1 second, this would probably results in a spatial scale of LEP events on the order of several hundred meters at the spacecraft altitude. I recommend the authors to add a brief discussion of these or similar straightforward inferences, and put them in the context of existing literature.
- **Response:** Thank you for this comment. To discuss the typical scale size of LEP patches, we direct the reviewer to Bortnik et al. 2006 JGR, figures 7 and 8:

[redacted]

In the limiting cases, where the spacecraft is at the edge (in longitude) of a precipitation patch, where the breadth is smallest (a few 100s km), it is possible to conceive that over the course of a few seconds the spacecraft completely traverses the spatial extent of the microburst. However, it is also possible for the spacecraft to observe the event while in the longitudinal center, and then due to the spacecraft velocity (about 7 km/s for LEO) and the large scale size (several 100s- few1000s km across), all variations in flux would likely be due to temporal variations. This could explain the large variability in shapes; the spacecraft location relative to the precipitation patch could determine whether a rising edge, falling edge, or neutral flux pattern is observed. We have added the following discussion to the text: “for example, based on simulation results from Bortnik et al 2006, the spacecraft may observe a larger spatial variation if it traversed an event near the edge of a precipitation patch as compared with at the center. Precipitation patches may vary dramatically over time and space, spanning on the order of 100s of km or fewer up to 1000s of km.”

Reviewer #2

- **R2C1:** In the second paragraph, Lines 54 through 60, there needs to be a reference to the only two papers that reported direct low altitude satellite measurements of LEP events, strongly correlated with electromagnetic waves (whistlers) from lightning: Inan, U. S., D. Piddiyachiy, W. B. Peter, J. A. Sauvaud, M. Parrot (2007), DEMETER satellite observations of lightning-induced electron precipitation, Geophys. Res. Lett., 34, L07103, doi:10.1029/2006GL029238. & Voss, H. D., et al. (1984), Lightning-induced electron precipitation, Nature, 312, 740–742, doi:10.1038/312740a0.
 - **Response:** Added as recommended.
- **R2C2:** In addition, in the context of the discussion of bouncing packets of electrons, and the asymmetry of the northern and southern mirror heights, there needs to be a reference to the paper that analyzed LEP events observed in the northern hemisphere in great detail: 219. Inan, U. S., M. Walt, H. Voss, and W. Imhof, Energy Spectra and Pitch Angle Distribution of Lightning-Induced Electron Precipitation: Analysis of an Event Observed on the S81-1 (SEEP) Satellite, J. Geophys. Res., 94, pp. 1379-1401, 1989.
 - **Response:** Thank you for pointing this out, we have added the following reference to the text: “While previous studies of LEP at lower energies have observed many events in the northern hemisphere [Inan], the relative dearth in our catalog is likely due to the lower flux of electrons at these higher energies.”

Reviewer #3

- **R3C1:** “Briefly, figure 5 shows the long-term dynamic of >2 MeV electrons correlated with the electromagnetic lightning waves induced microburst precipitations. But the demonstration of the link of MeV electron precipitations with lightning waves is made for >1 MeV electrons in the article, certainly not for >2MeV. This is totally misleading and not acceptable. I ask Fig 5 to be made for >1 MeV electrons, though I am finding a caveat with the >1 MeV SAMPEX channel reported in Selesnick et al. JGR 2015, which may prevent that approach to be a solution. Most of the data collected at energy threshold of 1 MeV are caused by lower energy electrons or high-energy protons (Selesnick et al. JGR 2015). As there is an issue with the >1 MeV data, the authors should say something about it.
 - **Response:** Thank you for this valuable insight. The Selesnick paper seems to comment on the PET instrument only, not the HILT instrument, which supplied the relevant data. We appreciate your comment on Figure 5, suggesting redoing the figure using HILT data only, and will be incorporating it into the manuscript.
- **R3C2:** Also, there is a second issue in this figure made from 10 years of measurements show geomagnetic activity in a very coarse way (1-month averaging). The events are plotted on top of the flux but cannot be well related to geomagnetic activity, which is not well visible at this scale. This second aspect is also misleading. That issue will be removed by a series of new distributions which I ask below the authors provide in order to properly relate their events to geomagnetic activity. Maybe >1 MeV flux can be replaced by other types of information (as slot filling event identification or the distributions I propose or other ideas I give below) or this figure should be removed.
 - **Response:** In-depth response to disagreements about Figure 5 are itemized below under their respective comments (R3C10-13).
- **R3C3:** The second disagreement is about Figure 6, in which we face here a coarse sketch of a dynamics which brings absolutely nothing to the article except a too caricatural, sometimes wrong, view of the radiation belt dynamics and of their projection onto the Earth’s map (fig 6a-b-c) as well as onto the projection of the Earth magnetic field (fig 6d). This is way too naïve. Not in Nature under my watch. The text associated with this figure is totally minor (copied below for your information). I simply recommend the removal of the text and the figure. This will not diminish the scientific achievements, which are mostly contained in Figure 1 to 4.
 - **Response:** Removed as recommended.
- **R3C4:** More corrections follow but these two reasons above are the reasons of my current rejection. I am not sure corrections can be done in a very short time as sometimes asked by Nature, particularly the correction about Figure 5. I recommend to give the authors the time they need to properly address them. Also, two new satellite missions will be in orbit in a couple of years or so to study more microburst events. I ask

the authors to mention them in the perspectives as their article can be inspiring and directly influence and impact new research studies. After appropriate corrections, I think this article will have the high potential and results to be considered for publication in Nature Communications.

- **Response:** Thank you for this encouragement, your thoughtful comments are appreciated.
- **R3C5:** The rarity of these event in the inner belt is linked to the absence of these MeV electrons in great majority. Please refer to Fennell et al. GRL 2015 on the upper limit of RB electrons and Baker et al. Nature 2015 on the “barrier” prohibiting MeV electrons to reach the inner belt. Please cite also, the rare intrusion of MeV electrons in the inner belt as discussed in Claudepierre et al. JGR 2019. These three references seem to me mandatory.
 - **Response:** Added the following, which was also part of the response to R1C7: “Given the relatively frequent occurrence of high-amperage lightning discharges, one may wonder why more LEP events were not observed over such a long epoch. There are a few potential explanations for this seeming contradiction, chief among which is the rarity of slot-filling events [Fennell, Baker, Claudepierre]. Other factors include the localization of the precipitation patch and the short timescale over which the precipitation patch propagates. Finally, the authored search algorithm was written to be as strict as possible, and so it is plausible that many more of these events occurred in the SAMPEX/HILT data than were identified.”
- **R3C6:** Abstract: “...where they do not reside long-term”. Unclear. Do you speak of the electrons seen in the microburst because they vanished after a short term or do you speak in general of radiation belt electrons? In the first case, you are not discussing the lifetime of these electrons; could you? Please rephrase to be clear.
 - **Response:** We have opted to remove this sentence, since it is ambiguous as you point out, and discuss the details of the lifetimes of MeV electrons in the manuscript (Figure 5 discussion).
- **R3C7:** “these results constitute the first direct evidence for lightning-driven precipitation of MeV electrons, illuminating a new perspective on the lifetimes and behavior of such high-energy particles so close to the Earth.”: this is indeed a very good result worth publishing in Nat. Comms. It is not only by lightning waves but by any VLF waves. I would like you mention that Cunningham et al. GRL 2020 could show precipitations of relativistic electrons by VLF-Transmitter waves up to 600-700 keV only, still above 450 keV found by Sauvaud et al. (2008), but not above 700 keV. The article is the first one to reach >1 MeV. That should support your claim even more.
 - **Response:** Thank you for your comment. This was changed to: “Relativistic electron precipitation by VLF waves has been previously observed, but the upper

limit for the energy spectra was 700 keV [Cunningham, Sauvaud]. Additionally, LEP of MeV electrons has been inferred from perturbations in subionospheric VLF waves [Inan], but these results constitute the first direct evidence for VLF-driven precipitation of MeV electrons in the inner belt and slot region, illuminating a new perspective on the lifetimes and behavior of such high-energy particles so close to the Earth.”

- **R3C8:** Please indicate in the text what is the altitude of SAMPEX.
 - **Response:** Added as recommended.
- **R3C9:** Please give the following distributions:
 - Distribution in duration of the events
 - Distribution in Nb of peaks of the events (if interesting and different from the duration, if not, then just consider a quick comment on this number).
 - Distribution with geomagnetic activity, maybe Kp, AE and Dst. Do you need to introduce “*” (starred indices) i.e the average over the previous 24h as sometimes done? Please choose according to the relevance you find.
 - Any other idea? See below the connection with slot filling events and how you may establish a correlation with them, shown by a distribution

Please try to enrich Fig 2 with the relevant distributions you could find. Please make fig 2 more meaningful and impactful. (As such it is great but a bit limited and we want to know more, plus add the information which is currently in Fig. 5, which cannot stay if showing >2MeV).

 - **Response:** Many of these distributions were added. Because the bounce period is fairly similar for most of these events, we choose to show a histogram of the number of peaks only, and just comment on the overall duration of events in the text (as the two are closely related). For a similar reason, we show the distribution in Dst in the manuscript but not AE or Kp, as the conclusions were similar: in general, these events take place during times of greater geomagnetic activity when compared with background distributions. We have included the latter two distributions below for the reviewer; in both cases, the values during the microburst events were greater than the baseline values.

To verify this statement statistically, a two-population t-test was used between the background Dst index and the distribution in Dst during the microburst events. The probability that these two samples are not statistically different is 0.46%, leading us to reject the null hypothesis and conclude that the microburst events do occur during geomagnetically disturbed times. For this reason, we conclude that the analysis is sufficient without introducing Dst*.

- Figure 2 of the manuscript has now been updated based on the above, and caption and description updated as well to discuss these new panels (see lines 107-114 of the tracked changes manuscript).

➤ **R3C10:** Fig 5 shows >2MeV electrons while here the authors use >1 MeV electrons. This is not possible because it suggests to the reader the precipitation are occurring at >2MeV, way above the natural limit of inner belt electrons. Fig 5 should plot the energy channel which is used in the article and not a different one. Now, it is unfortunate that the main >1 MeV channel is likely not usable as found by Selesnick in 2015.... I do not think

this issue can be put under the rug. The corruption of the >1 MeV channel is though—I think—fully unrelated with the microburst detector. Please discuss that once the results of Selesnick et al. 2015 will be cited.

- **Response:** We appreciate this comment - you make a great point about the PET instrument MeV measurements (as detailed in Selesnick et al.) To address this, we have updated Figure 5 to now include only HILT instrument >1 MeV measurements rather than PET >2 MeV.
- **R3C11:** Figure 5, which is made from 10 years of measurements, shows geomagnetic activity in a very compressed way due to the packing of the years. The resolution of Dst is very coarse: an averaged value over 1 month. The events are plotted on top of the flux but cannot be well related to local geomagnetic activity, which is not well visible at this scale. This second aspect is also misleading. I think the distribution with Dst I asked above will be enough to relate one with the other, certainly more precise.
- **Response:** We have now updated the Dst resolution to be daily rather than monthly averaged for this figure. We have also included a histogram demonstrating the distribution of these events in Dst to Figure 2; hopefully this will provide insight into the general trend of Dst index with respect to these events.
- **R3C12:** “ From the figure, we can see that the microburst events tend to follow closely after large dips in Dst.” I cannot conclude that from the fig.5 due to the Dst resolution.
- **Response:** As mentioned in R3C11, we have adjusted the Dst resolution to be daily rather than monthly, and have added stars where large dips in Dst (<-50 nT) are observed.

- **R3C13:** “Figure 5 shows that many of the microburst events occurred during or following periods of increased geomagnetic activity, while the slot region was filled with MeV electrons.” I do not see that from the figure. Can you make a better demonstration? Can you relate or correlate slot filling events with these events? See Turner et al. 2015. That may work. But this will only say that the presence of these electrons in the inner belt is rare enough that they have to be brought during active time, which I do not contest. As the authors, I think high activity is required to have the presence of these electrons in the inner belt in the first place. But, then, what is happening? Do we have the situation that

any of the many lightning waves will cause their instantaneous or rapid precipitations? Or, these electrons stay some time trapped until they are hit by a lightning wave, totally independently of the geomagnetic activity? The authors are suggesting the first case based on a crude comparison I contest. Please make a better work at relating activity with these events. For instance, what is the time between the event and the closest significant storm which you think brought these electrons? ... how would be that distribution for the 45 events? As you do, you may find a better way to demonstrate your claim, without the use of the current fig. 5.

- **Response:** Thank you for this suggestion. To better relate the timing of these events with geomagnetic disturbances, we have added the following plot to Figure 5 (as a part **b** to the figure in R3C12), which shows the delay time between microburst events and geomagnetic storms, as identified as periods with $Dst < -50$ nT following Turner et al. 2015. It seems that most of the events occur within 5 days of the dip in Dst, and nearly all occur within 15 days (roughly 2 weeks). In comparison, we also show the distribution in spacing between geomagnetic storms in general, as a baseline, which can be up to half a year or longer. Hopefully these distributions better illustrate that microbursts occur relatively soon after geomagnetic storms.

- **R3C14:** I am simply against Fig 6 by all means. In a-b-c the dynamics is caricatural. In fig 6d, the spread of the radiation belts is way too rough. This figure is an introduction figure to maybe a course and even I found it too inaccurate to show it to students. The SAA is giant, the radiation belts are too wide for the North one particularly, etc. If drawn by Picasso, maybe.... There is no particular message in this figure. The storm makes a relativistic radiation belt appearing in c) gone in a) and b). Where did you see that? This figure is too specula/ve and naïve. I am recommending not to publish that figure. The article is perfectly fine without. Looking at the text referring to this figure in the concluding paragraph, copied below and colored in blue, it is very weak and can simply be removed:

“The combination of numerous disparate phenomena are needed to produce the spatial and temporal distributions of MeV LEP events revealed here. Figure 6 highlights this combination, with Figures 6a-c demonstrating the dependence in time (i.e. during or after geomagnetic storms), and Figure 6d showing the dependence in space (i.e. the overlapping of the inner radiation belts, lightning distributions, and the South Atlantic Anomaly). Accounting for and modelling the behavior of energetic particles in the near-Earth space environment remains an ongoing challenge, particularly for the space industry, where these particles can damage space equipment or even fatally irradiate

humans [30]. These results may aid in this challenge, and can form the basis for more detailed analysis into the temporal behavior of MeV electrons in the inner radiation belt. In addition, the confluence of Earth-generated processes, such as lightning; near-Earth space phenomena, such as microbursts; and solar activity, such as slot-filling geomagnetic storms; serves as a reminder of the connection between these seemingly unrelated regions.”

With the removal I ask, this last paragraph would become perfectly fine with me:

“Accounting for and modelling the behavior of energetic particles in the near-Earth space environment remains an ongoing challenge, particularly for the space industry, where these particles can damage space equipment or even fatally irradiate humans [30]. These results may aid in this challenge, and can form the basis for more detailed analysis into the temporal behavior of MeV electrons in the inner radiation belt. In addition, the confluence of Earth-generated processes, such as lightning; near-Earth space phenomena, such as microbursts; and solar activity, such as slot-filling geomagnetic storms; serves as a reminder of the connection between these seemingly unrelated regions.”

That showed how minor that text was. See below where I propose the authors more perspectives.

- **Response:** Removed the figure and text as recommended.
- **R3C15:** Lightning strikes can be tracked with the WWLLN lightning detection network as explained in the text. As indicated by the authors, at the time their data were collected, this network was unavailable. This causes the authors to limit their analysis in the US region due to the NLDN network coverage, which accuracy is also lower over the Atlantic region, where the authors have many events. This is an unavoidable caveat. However, for new upcoming missions, which are also targeting microburst events, such as IMPAX (led by Minnesota Univ.) and the Canadian RADICALS missions, this network should provide a good way to identify accurately the location of the lightning source of lightning related microbursts as well as enlarge the region of study to the whole world, instead of the US only. Your article is an invitation to do so. Therefore, I would like you take advantage of this argument to make your article more impactful in directly citing in the perspectives the possibility of extending your results with the upcoming new measurements from IMPAX or RADICALS, themselves being possibly related to new lightning network such as WWLLN (see for instance in Ripoll et al. GRL 2019) for a world-wide assessment.
 - **Response:** Added the following: “**New high-time resolution measurements of energetic electrons from the inner radiation belts such as those from the upcoming IMPAX [Colpitts] or RADICALS [Mann] missions could take advantage of the worldwide lightning network currently available and provide a more**

complete, global picture of the relationship between lightning and microbursts in the inner radiation belt.”

- **R3C16:** In Fig 4, try neon yellow for lightning, not blue because one lightning in the Florida see is not visible.
 - **Response:** Revised as recommended.
- **R3C17:** Fig 6a: “an asymmetrical distribution of energetic particles about the Earth”. Rephrase. Fig6d: not commented in the legend. Though I recommend to remove that figure.
 - **Response:** Understood, this figure has been removed.
- **R3C18:** Please change
“ lightning discharges generate can more easily propagate into the magnetosphere on the nightside, producing a prevalence of LEP on the nightside only [21–23].” By
“lightning discharges generate can more easily propagate into the magnetosphere on the nightside [21–23], producing a prevalence of LEP on the nightside only.” (unless these articles have a specific aspect on LEP but I don’t think so. Still, if so, you can break the citation in two pieces, one relative to wave observations, the other to LEP)
 - **Response:** Revised as recommended.

REVIEWER COMMENTS

Reviewer #1 (Remarks to the Author):

Second review report by Ondrej Santolik

on Manuscript#: NCOMMS-23-62997-T "Lightning-induced relativistic electron precipitation from the inner radiation belt"

submitted to Nature Communications by Max Feinland et al.

I read through the responses, revised text, and figures and my opinion is that the authors made a great effort toward responding to the comments of the reviewers. I still recommend the paper for publication. I have just two minor comments concerning the responses to my previous suggestions, which the authors may consider for the final version of the paper.

ad R1C3: It seems to me that the average time separation of the microburst peaks can be determined with a lower uncertainty than the sampling interval. Dividing the duration of the sequence by the number of peaks (between 4 and 12, according to the newly added histogram in Fig.2) should do the job, unless I misunderstood something relevant in the description of the data. I understand that the authors plan to work on a better characterization of the peaks in a follow-up study but if they agree with this suggestion it might be instructive to see a histogram of the resulting average peak separations as an additional panel in Fig 2. An additional plot or at least a sentence in the text may then describe a range of possible energies, similar to the example in the response.

ad R1C5: I suggest to add a plot of the nice sister events from 30 January 2003 to the paper as Supplementary Information, in support of the newly added text, which is now in the revised version of the paper.

Reviewer #1 (Remarks on code availability):

The code is represented by a matlab script containerized in a web based system of the codeocean.com company. The script contains matlab functions for detection, reporting, and plotting of sequences of microbursts. The code itself contains useful comments and instructions. I didn't try to run it but I went through the code and I think that it contains all necessary details about the data analysis reported in the text of paper.

June 2 2024

Object: Review of NCOMMS-23-62997-T

Dear Editor,

I reviewed the revised version of "Lightning-induced relativistic electron precipitation from the inner radiation belt" by Max Feinland et al. submitted to Nature Communications.

I had indicated rejection with encouragement to resubmit because the article results were indeed very encouraging but some points had to be removed or greatly corrected, with the risk for the authors not to be able to do it. The authors have integrated and answered all my comments, in a great manner. Specifically, they have:

- clarify a big doubt I had on the electron flux measurements corruption and removed any data which could have been coming from a corrupted instrument of the SAMPEX satellite. Now, I trust the measurements they use.
- added the distributions I had asked and commented more on geomagnetic activity, improving the event description.
- re-generated their figure 5 with >1 MeV data (and not >2 MeV as originally submitted) which was hurting badly their results/conclusions. I was afraid the >1 MeV would not be available or corrupted and that was a big issue for this article. I am glad the authors could have the correct measurements.
- removed their former fig 6 and related text, which I was strongly against. All good now.
- added the references I had asked on related work
- removed statements that were not fully justified

The modification of the text from the other reviewers' comments, plus the improvement of writing and precision of the results make this article close to ready for publication. This will be a great article on the capability of lightning strokes for precipitating relativistic electrons. I found minor points below to correct, listed below, with 2 of them to try to improve the readability of the figure 2 and 5. Once this is done, I am accepting the article for publication in Nature Communications.

Best regards

Minor corrections:

Figure 2d: The legend is "d, Distribution in Dst index for microburst events (red), as compared to the Dst index over all time surveyed (blue). The Dst index during the microburst events is in general lower. ". There is no red. There are 2 brown actually, which makes it 3 colors with one which is not defined. Please correct. Also Figure 5b has a similar issue. I guess it is the superposition of brown/orange with blue giving dark brown. Just make it clear. Like if you use black you won't have a third confusing color. Or remove some transparency.

I don't understand that sentence "The first distribution skews markedly shorter than the second. " in the legend. Do you mean all events falling in the first bin (0 to 1 day are actually falling within a second of a storm?). Unclear. Please rephrase. Plus this point is not discussed in the text (or I missed it?).

« during statistically lower Dst indices » : can you add '(corresponding to high activity)' just for unfamiliar readers to know. Can you be a little more precise and give an idea of the Dst value (like a mean value)? Something like "(Dst_mean=XXX or Dst ~XXX, corresponding to high activity)". Maybe this is already said in the text with ("Dstmin \leq -50 nT "). Then just write that value.

Also earlier in the text : " Dst index during an electron microburst event was, on average, lower than the baseline number. " What does that mean? What is the baseline number?

Figure 5: there is a visual artefact. You have black lines coming from low flux values (even maybe no data or NaN) which compete with the Dst line and blurs its visualization. Can you try to do something, like either change that background color (white?) or change the color of the Dst line? Maybe a Dst in magenta (as the star symbol)? Or/and you round the low/NaN flux data to some low flux level in the blue?

In "...RADICALS [43] missions could take advantage of the worldwide lightning networks...". At the beginning of the article you removed the information that WWLLN was unavailable during the study. So in this sentence we don't understand why WWLLN is showing up now. You want to write something like: "...RADICALS [43] missions could take advantage of the worldwide lightning networks (unavailable during SAMPEX mission)...". or maybe "...networks (unavailable for the dates of the events considered here)...". Then we understand you project toward new missions and new lightning data. Other lightning network could be used too : "...could take advantage of the XXXX as well as the worldwide lightning..."

Line 413: typo . coloredModeling

(1 MeV, i.e. 0.94c) : is c defined?

Author Responses to Reviewer Comments

“Lightning-induced relativistic electron precipitation from the inner radiation belt”

Manuscript submitted to Nature Communications by Feinland et al.

We thank the reviewers for their thoughtful contributions and suggestions and we have addressed each comment below.

Author responses are in **black**. Changes to the manuscript text are in **red**.

Reviewer #1

- **ad R1C3:** It seems to me that the average time separation of the microburst peaks can be determined with a lower uncertainty than the sampling interval. Dividing the duration of the sequence by the number of peaks (between 4 and 12, according to the newly added histogram in Fig.2) should do the job, unless I misunderstood something relevant in the description of the data. I understand that the authors plan to work on a better characterization of the peaks in a follow-up study but if they agree with this suggestion it might be instructive to see a histogram of the resulting average peak separations as an additional panel in Fig 2. An additional plot or et least a sentence in the text may then describe a range of possible energies, similar to the example in the response.
 - **Response:** We have added the distribution of minimum peak separations to the histograms in Figure 2, as shown in panel **d** below.

- We furthermore added the following to the text: “Figure 2d shows the minimum spacing between peaks for each event; the minimum was used because the peak separation is expected to increase in time as the distribution in electron energies produces a “spreading” of the peaks (Shumko et al. 2018, Voss et al. 1998). Assuming a dipole model as described by Schulz and Lanzerotti (1974), the characteristic electron energy was calculated for each event. The calculated energies range from 323 keV to 7.81 MeV. However, since the HILT time resolution is 20 ms, and the bounce period around $L = 2$ is approximately 200ms, there can be quite a bit of uncertainty in the calculated bounce period and therefore electron energy. For example, for an observed bounce period of 200 ms at $L = 2$ and equatorial pitch angle of 15 degrees, the estimated energy would be 1.4 MeV, whereas a bounce period of 220 ms at the same L and pitch angle would be 550 keV.”
- **ad R1C5:** I suggest to add a plot of the nice sister events from 30 January 2003 to the paper as Supplementary Information, in support of the newly added text, which is now in the revised version of the paper.
 - **Response:** Thank you for this suggestion, this was incorporated.

Reviewer #2

Reviewer #2 did not have any additional comments in this round of review.

Reviewer #3

- Figure 2d: The legend is “d, Distribution in Dst index for microburst events (red), as compared to the Dst index over all time surveyed (blue). The Dst index during the microburst events is in general lower. ”. There is no red. There 2 brown actually, which makes it 3 colors with one which is not defined. Please correct. Also Figure 5b has a similar issue. I guess it is the superposition of brown/orange with blue giving dark brown. Just make it clear. Like if you use black you wont have a third confusing color. Or remove some transparency.
 - **Response:** The figure colors have been modified as suggested, and the text was modified to account for this: “d, Distribution in Dst index for microburst events (red), as compared to the Dst index over all time surveyed (blue). **The overlap between these quantities is shown in purple.** The Dst index during the microburst events is in general lower.” Please see plot shown in the response to Reviewer #1.
- I don’t understand that sentence “The first distribution skews markedly shorter than the second. ” in the legend. Do you mean all events falling in the first bin (0 to 1 day are

actually falling within a second of a storm?). Unclear. Please rephrase. Plus this point is not discussed in the text (or I missed it?).

- **Response:** We apologize that this sentence was unclear. This was rephrased to: “The distribution of delay times between geomagnetic storms and microburst events is markedly shorter on average than the delay between geomagnetic storms.” This is discussed more in the text at lines 196-200 of the newly submitted manuscript (tracked changes version).
- « during statistically lower Dst indices » : can you add ‘(corresponding to high activity)’ just for unfamiliar readers to know. Can you be a little more precise and give an idea of the Dst value (like a mean value)? Something like “(Dst_mean=XXX or Dst ~XXX, corresponding to high activity)’. Maybe this is already said in the text with (“Dstmin ≤ -50 nT ”). Then just write that value.
 - **Response:** Great suggestion, the text was modified to: “... the microburst events occurred during statistically lower Dst indices (corresponding to higher geomagnetic activity). The mean Dst index during microburst events was -26.73 nT, while the mean Dst for all time surveyed was -16.40 nT.”
- Also earlier in the text :” Dst index during an electron microburst event was, on average, lower than the baseline number. ” What does that mean? What is the baseline number?
 - **Response:** We apologize for the lack of clarity. The text was rephrased to: “The average Dst index during the electron microburst events was lower than the average Dst index during the full 10 years surveyed.”
- Figure 5: there is a visual artefact. You have black lines coming from low flux values (even maybe no data or NaN) which compete with the Dst line and blurs its visualization. Can you try to do something, like either change that background color (white?) or change the color of the Dst line? Maybe a Dst in magenta (as the star symbol)? Or/and you round the low/NaN flux data to some low flux level in the blue?
 - **Response:** We attempted to remove the low/NaN flux data so that the Dst line is clearer. We also changed the line style of the Dst index to be continuous in an attempt to improve clarity and slightly modified the colorbar. We hope that the plot reads more easily now.

- We also slightly changed the color of the bars in the histogram for Figure 5b to be consistent with the colors used in Figure 2d. The revised figure is shown below.

- In “...RADICALS [43] missions could take advantage of the worldwide lightning networks...”. At the beginning of the article you removed the information that WWLLN was unavailable during the study. So in this sentence we don’t understand why WWLLN is showing up now. You want to write something like: “...RADICALS [43] missions could take advantage of the worldwide lightning networks (unavailable during SAMPEX mission)...”. Or maybe “...networks (unavailable for the dates of the events considered here)...”. Then we understand you project toward new missions and new lightning data. Other lightning network could be used too : “...could take advantage of the XXXX as well as the worldwide lightning...”

- **Response:** Thank you for pointing out this oversight, the text was changed to “... RADICALS [Mann] missions could take advantage of the worldwide lightning networks (which were not readily available during the time surveyed in this study) ...”

- Line 413: typo . coloredModeling

- **Response:** This was fixed, thank you for pointing it out.

- (1 MeV, i.e. 0.94c) : is c defined?

- **Response:** Changed text to “1 MeV, i.e. 0.94c where c is the speed of light”

REVIEWERS' COMMENTS

Reviewer #1 (Remarks to the Author):

Third review report by Ondrej Santolik

on Manuscript#: NCOMMS-23-62997B "Lightning-induced relativistic electron precipitation from the inner radiation belt"

submitted to Nature Communications by Max Feinland et al.

The authors again did a great job in responding to the comments of the reviewers. I have no additional comments. I recommend the paper for publication.

Reviewer #1 (Remarks on code availability):

The code is represented by a matlab script containerized in a web based system of the codeocean.com company. The script contains matlab functions for detection, reporting, and plotting of sequences of microbursts. The code itself contains useful comments and instructions. I didn't try to run it but I went through the code and I think that it contains all necessary details about the data analysis reported in the text of paper.

July 8 2024

Object: Review of NCOMMS-23-62997-T, third round

Dear Editor,

I reviewed the last revised version of “Lightning-induced relativistic electron precipitation from the inner radiation belt” by Max Feinland et al. submitted to Nature Communications.

Last round of review I had indicated some remaining corrections, in particular to improve the figures and to precise some last unclarities left in the text. This has been fully done and the article is in its final form for me.

As I have read the article once again, I thought that in the discussion section, there could be a short comment added about precipitations which could be caused by super strong lightning events, called superbolts (Turman, 1977), and that would be interesting to know if the related electron precipitation would be noticeably stronger. I am asking the authors to add a sentence about that, giving an example on how to below, knowing there was a recent Nature Communications article on this topic (Ripoll et al., 2021), which would be worth citing.

As such, the Feynland et al. article is a well-documented and written article on the capability of lightning strokes to precipitate relativistic electrons, which allows me to accept the article for publication in Nature Communications.

Best regards

Minor addition in the discussion section: as a perspective, the authors could add, for instance, after

‘They also suggest a spatial variability of LEP microbursts previously uncharacterized by former studies; this high temporal resolution, in-situ data allows for an in-depth look at the nature of these events, and reveals details of the wave-particle interaction process that were unable to be captured by previous studies.’

One sentence like:

Extreme lightning strokes, *superbolts* (Turman, 1977), radiate strong electromagnetic waves (Ripoll et al., 2021) and could, in principle, cause stronger LEP than common lightning events, which would be interesting to observe in order to better characterize the nature of wave-particle interactions.

(The authors please feel free to rephrase as needed).

Turman, B. N. Detection of lightning superbolts. J. Geophys. Res. 82, 2566–2568 (1977).

Ripoll J-F, Farges T, Malaspina DM, Lay EH, Cunningham GS, Hospodarsky GB, et al. Electromagnetic Power of Lightning Superbolts from Earth to Space. Nat Commun (2021) 12:3553. doi:10.1038/s41467-021-23740-6

Author Responses to Reviewer Comments

“Lightning-induced relativistic electron precipitation from the inner radiation belt”

Manuscript submitted to Nature Communications by Feinland et al.

Reviewer #3

- Minor addition in the discussion section: as a perspective, the authors could add, for instance, after
'They also suggest a spatial variability of LEP microbursts previously uncharacterized by former studies; this high temporal resolution, in-situ data allows for an in-depth look at the nature of these events, and reveals details of the wave-particle interaction process that were unable to be captured by previous studies.'
One sentence like:
Extreme lightning strokes, superbolts (Turman, 1977), radiate strong electromagnetic waves (Ripoll et al., 2021) and could, in principle, cause stronger LEP than common lightning events, which would be interesting to observe in order to better characterize the nature of wave-particle interactions.
(The authors please feel free to rephrase as needed).
- **Response:** We added the following sentence to the text: “Extreme lightning bolts, also known as superbolts (Turman 1977), radiate strong electromagnetic waves (Ripoll 2021) and could, in principle, cause stronger LEP than common lightning events. Observations of such lightning discharges and the associated electron precipitation could facilitate better characterization of the nature of wave-particle interactions.”